# Central Sleep Apnea Is Associated with an Abnormal P-Wave Terminal Force in Lead V_1_ in Patients with Acute Myocardial Infarction Independent from Ventricular Function

**DOI:** 10.3390/jcm10235555

**Published:** 2021-11-26

**Authors:** Jan Pec, Michael Wester, Christoph Fisser, Kurt Debl, Okka W. Hamer, Florian Poschenrieder, Stefan Buchner, Lars S. Maier, Michael Arzt, Stefan Wagner

**Affiliations:** 1University Heart Center Regensburg, University Hospital Regensburg, 93053 Regensburg, Germany; jan.pec@klinik.uni-regensburg.de (J.P.); michael.wester@ukr.de (M.W.); christoph.fisser@ukr.de (C.F.); kurt.debl@ukr.de (K.D.); Lars.Maier@klinik.uni-regensburg.de (L.S.M.); Michael.Arzt@klinik.uni-regensburg.de (M.A.); 2Department of Radiology, University Hospital Regensburg, 93053 Regensburg, Germany; Okka.Hamer@klinik.uni-regensburg.de (O.W.H.); Florian.Poschenrieder@klinik.uni-regensburg.de (F.P.); 3Department of Internal Medicine, Cham Hospital, 93413 Cham, Germany; stefan.buchner@sana.de

**Keywords:** acute myocardial infarction, p wave terminal force, central sleep apnea, sleep-disordered breathing

## Abstract

Sleep-disordered breathing (SDB) is highly prevalent in patients with cardiovascular disease. We have recently shown that an elevation of the electrocardiographic (ECG) parameter P wave terminal force in lead V_1_ (PTFV_1_) is linked to atrial proarrhythmic activity by stimulation of reactive oxygen species (ROS)-dependent pathways. Since SDB leads to increased ROS generation, we aimed to investigate the relationship between SDB-related hypoxia and PTFV_1_ in patients with first-time acute myocardial infarction (AMI). We examined 56 patients with first-time AMI. PTFV_1_ was analyzed in 12-lead ECGs and defined as abnormal when ≥4000 µV*ms. Polysomnography (PSG) to assess SDB was performed within 3–5 days after AMI. SDB was defined by an apnea-hypopnea-index (AHI) >15/h. The multivariable regression analysis showed a significant association between SDB-related hypoxia and the magnitude of PTFV_1_ independent from other relevant clinical co-factors. Interestingly, this association was mainly driven by central but not obstructive apnea events. Additionally, abnormal PTFV_1_ was associated with SDB severity (as measured by AHI, B 21.495; CI [10.872 to 32.118]; *p* < 0.001), suggesting that ECG may help identify patients suitable for SDB screening. Hypoxia as a consequence of central sleep apnea may result in atrial electrical remodeling measured by abnormal PTFV_1_ in patients with first-time AMI independent of ventricular function. The PTFV_1_ may be used as a clinical marker for increased SDB risk in cardiovascular patients.

## 1. Introduction

Sleep-disordered breathing (SDB) is a common co-morbidity in patients with cardiovascular disease [1,2,3]. Nearly 50% of patients undergoing coronary artery bypass surgery (CABG) were found to have SDB [4]. Obstructive sleep apnea (OSA) is characterized by the presence of repetitive episodes of upper airway collapse. In contrast, central sleep apnea (CSA) is caused by an intermittent lack of centrally controlled respiratory drive, which often manifests as Cheyne–Stokes respiration and leads to significant oxygen desaturation. Epidemiologic studies indicate a strong association between both OSA and CSA and atrial fibrillation (AF) [5,6]. The most commonly used treatment is continuous positive airway pressure (CPAP), which can alleviate the clinical symptoms of SDB. However, the adherence to this therapy is generally poor and no significant benefit has been shown regarding cardiovascular outcome in patients with OSA [7,8]. The recent randomized controlled trial led by Traaen et al. demonstrated that CPAP treatment does not affect the burden of AF after 5 months of therapy [9]. Moreover, adaptive servo-ventilation has even been reported to increase the risk of cardiovascular death in patients with reduced left ventricular ejection fraction (LV EF) and CSA [10]. Therefore, identification of novel risk markers and new treatment options are of utmost importance.

The P wave terminal force in electrocardiographic (ECG)-lead V_1_ (PTFV_1_) was firstly introduced by Morris et al. in 1964 [11]. It is defined as the algebraic product of the amplitude and duration (µV*ms) of the negative area of the P-wave in lead V_1_ (Figure 1). Accumulating evidence has since linked an abnormally large PTFV_1_ to atrial dysfunction [4] and AF [12] with increased risk for cardioembolic or cryptogenic stroke [13,14]. Moreover, an abnormally PTFV_1_ has also been shown to predict cardiovascular risk and cardiac death or hospitalization for heart failure in patients with prior myocardial infarction [15].

Interestingly, we have recently shown that an abnormally large PTFV_1_ was associated with atrial functional and electrical remodeling by activation of Ca/calmodulin-dependent protein kinase II (CaMKII). CaMKII-dependent dysregulation of cardiomyocytes ion homeostasis has already been associated with atrial pathologies [16], and increased CaMKII-dependent atrial pro-arrhythmic activity was found in cardiovascular patients with SDB [4]. Since CaMKII can be activated by oxidation, intermittent hypoxia could be an important upstream factor.

To date, however, it is unclear which pathophysiologic factor—be it negative intrathoracic pressure fluctuations, intermittent hypoxia, increased production of reactive oxygen-species (ROS), or autonomic imbalance [17]—might be most significant for atrial electrical remodeling. In addition, little is known about the relationship between PTFV_1_ and SDB in patients with acute myocardial infarction. Therefore, this present study investigated the relationship between SDB and SDB-related hypoxia with PTFV_1_ in patients presenting with acute myocardial infarction.

## 2. Materials and Methods

### 2.1. Study Approval and Design

We performed a sub-analysis of a prospective observational study in patients with acute MI that were enrolled at the University Medical Center Regensburg (Regensburg, Germany) between March 2009 and March 2012. Details of the study design have been published previously [3].

Patients (age 18–80 years) with a first-time AMI and successful percutaneous coronary intervention (PCI) treated at the University Hospital Regensburg within 24 h after symptom onset were eligible for inclusion. Exclusion criteria were previous MI or previous PCI, indication for surgical myocardial revascularization, cardiogenic shock, contraindications for cardiac magnetic resonance imaging (CMR), and severe comorbidities (e.g., lung disease, stroke, treated SDB). The study protocol was reviewed and approved by the local institutional ethics committee (Regensburg, 08-151) and is in accordance with the Declaration of Helsinki and Good Clinical Practice. A written informed consent was obtained from all patients prior to enrolment.

Of 252 consecutive patients who underwent percutaneous coronary intervention, 74 patients were eligible for the prospective observational study, which involved an evaluation of cardiac function (CMR) and SDB severity at the time of MI. In total, 34 patients were excluded from this sub-analysis due to missing CMR (*n* = 10), missing polysomnography (*n* = 6), and atrial fibrillation (*n* = 2). The final sub-analysis included 56 patients, who were divided into two cohorts depending on the PTFV_1_ (PTFV_1_ < 4000 µV*ms [*n* = 40] and PTFV_1_ ≥4000 µV*ms [*n* = 16]) (Figure 2).

### 2.2. Electrocardiography

Standard 12-lead electrocardiograms were recorded at a paper speed of 50 mm/s and a standardization of 10 mm/1 mV. All ECGs were digitally processed and scaled using ImageJ (Version 2.00; Java-based image processing program; LOCI, University of Wisconsin, USA) and individually analyzed by two skilled physicians (mean of 3 consecutive P waves). Both investigators were blinded to the clinical and MRI data. PTFV_1_ was defined as the algebraic product of amplitude (µV) and duration (ms) of the terminal negative component of the P wave in lead V_1_ (Figure 1) also known as Morris-Index [11]. A PTFV_1_ of ≥ 4000 µV*ms was considered to be abnormal.

### 2.3. Polysomnography

Polysomnography (PSG) was performed in all subjects using standard polysomnographic techniques (Alice System; Respironics, Pittsburgh, PA, USA) as previously described [3]. Briefly, respiratory efforts were measured with the use of respiratory inductance plethysmography and airflow by nasal pressure. Sleep stages and arousals, as well as apneas, hypopneas, and respiratory effort-related arousals, were determined according to the American Academy of Sleep Medicine guidelines [18] by an experienced sleep technician blinded to the clinical data. Hypopneas were classified as obstructive if there was out-of-phase motion of the ribcage and abdomen, or if airflow limitation was present. In order to achieve optimal distinction between obstructive and central hypopneas without using an esophageal balloon, we used additional criteria, such as flattening, snoring, paradoxical effort movements, arousal position relative to hypopneas, and associated sleep stage (rapid eye movement (REM)/non-REM). SDB was defined by an apnea-hypopnea-index (AHI) > 15/h determined as the number of central or obstructive apnea and hypopnea episodes per hour of sleep. CSA was defined as >50% central apneas and hypopneas of all apneas and hypopneas. Pulse oximetry implemented in PSG was used to measure oxygen saturation and ODI (number of events per hour in which oxygen saturation decreased by ≥3% from baseline).

### 2.4. Cardiovascular Magnetic Resonance

Details of CMR data acquisition have been previously described [3]. Shortly, CMR studies were performed on a clinical 1.5 Tesla scanner (Avanto, Siemens Healthcare Sector, Erlangen, Germany) using a phased array receiver coil during breath-hold and that was ECG triggered. Examination of ventricular function was performed by acquisition of steady-state free precession (SSFP) cine images in standard short axis planes (trueFISP; slice thickness 8 mm, inter-slice gap 2 mm, repetition time 60.06 ms, echo time 1.16 ms, flip angle 60°, matrix size 134 × 192, and readout pixel bandwidth 930 Hz*pixel^−1^). The number of Fourier lines per heartbeat was adjusted to allow the acquisition of 25 cardiac phases covering systole and diastole within a cardiac cycle. The field of view was 300 mm on average and was adapted to the size of the patient. Calculation of left ventricular volumes and EF was performed in the serial short axis slices using commercially available software (syngo Argus, version B15; Siemens Healthcare Sector).

### 2.5. Statistical Analysis

Continuous variables were compared by Student’s T-test or Welch’s Test depending on their variance. The Chi-square or Fisher’s exact test were used for categorial variables depending on the number of observations. Continuous variables are expressed as mean ±95% confidence interval (CI), and categorial variables as frequencies and percentages, respectively. After linear regression of PTFV_1_ or AHI with important clinical factors, multivariate linear regression was performed for all variables with a *p* value < 0.2. An intra class correlation (ICC, by two-way mixed model, type absolute agreement) was used to assess the reproducibility of PTFV_1_ analysis. All reported P values are two-sided and the threshold for significance was set at *p* < 0.05. Statistical analysis was performed in SPSS (SPSS Statistics for Mac OS, Version 26.0 Armonk, NY, USA: IBM Corp.).

## 3. Results

### 3.1. Study Population

A total of 56 patients consisting of 80% men with an age of 55 ± 9.9 years were separated into groups with normal and abnormal PTFV1 (baseline characteristics in Table 1). There was no significant difference in demographic parameters or comorbidities, such as age, gender, arterial hypertension, diabetes mellitus, hypercholesterolemia, or smoking.

Patients with abnormal PTFV_1_ presented significantly less with ST segment elevation myocardial infarction (STEMI) (*p* = 0.035) and had higher levels of NT-proBNP at discharge (*p* = 0.002) (Table 1). The LV EF was mildly reduced in both groups but worse in patients with abnormal PTFV_1_ (43.15 ± 11.51% vs. 48.93 ± 7.45%, *p* = 0.035). Interestingly, volumetric parameters for LA size and function, such as LA fractional area change (FAC) or systolic LA area, were not significantly increased in patients with abnormal PTFV_1_ (Table 1), indicating that the magnitude of PTFV_1_ more likely reflects electrical but not structural remodeling as published previously [19].

### 3.2. Central Sleep Apnea Is Independently Associated with Abnormal PTFV_1_

Respiratory and sleep characteristics are shown in Table 2. The Epworth Sleepiness Scale score reflecting the daytime sleepiness was within the normal range in both groups (Table 2). Interestingly, in patients with abnormal PTFV_1_, SDB was highly prevalent (86.7%), with significantly more patients exhibiting central but not obstructive sleep apnea (Table 2). In contrast, only a minority of patients with normal PTFV_1_ had SDB (42.5%) and if so, a majority was obstructive (Table 2). Moreover, central (cAHI) but not obstructive (oAHI) apnea events were significantly associated with the magnitude of PTFV_1_ (Table 3). Importantly, the extent of oxygen desaturation (ODI) was an even stronger predictor of the extent of PTFV_1_ than that of the frequency of central apneas (R^2^ = 0.268, Table 3). In contrast to this association, the mean arterial oxygen saturation was similar in both groups. There was a trend towards lower minimum arterial oxygen saturation in the group with patients with abnormal PTFV_1_ (85.74 ± 5.87 vs. 82.20 ± 6.09, *p* = 0.055) (Table 2).

To test for possible confounding, multivariate linear regression was performed. The association of both ODI and cAHI with the magnitude of PTFV_1_ remained significant after inclusion of important co-factors, such as age, LVEF, eGFR, and NT-proBNP at discharge. Importantly, the associations of both ODI and cAHI were also independent from obstructive apnea events. For cAHI, R^2^ was 0.256 (adj. R^2^ = 0.186; *p* = 0.014, Table 4), and for ODI, R^2^ was 0.408 (adj. R^2^ = 0.317; *p* = 0.002, Table 4).

### 3.3. PTFV_1_ as a Diagnostic Marker for Predicting Sleep-Disordered Breathing

Univariate linear regression for AHI indicated that beside PTFV_1_, BMI, NT-proBNP at discharge, systolic LA area, LVEF, and smoking status may correlate with apnea and hypopnea events. Strikingly, after incorporation of these factors into a multivariate linear regression model, only PTFV_1_ significantly correlated with the magnitude of AHI (model 1, R^2^ = 0.326 (adj. R^2^ = 0.213); *p* = 0.021, Table 5). Similarly, after dichotomizing PTFV_1_ into normal and abnormal, the presence of an abnormal PTFV_1_ significantly predicted a more severe AHI in multivariate linear regression (model 2, B 21.495; CI [9.097, 20.193]; *p* < 0.001, Table 5).

Interestingly, no meaningful interactions were found with myocardial ischemia markers, such as troponin I or creatine kinase and abnormal PTFV_1_ (Table 5), despite the higher prevalence of STEMI in the group with normal PTFV_1_ (92.5% vs. 68.8%).

## 4. Discussion

In the present study, we investigated the relationship between SDB and SDB-related hypoxia with PTFV_1_ in patients presenting with acute myocardial infarction.

We show here that nocturnal oxygen desaturation in SDB was associated with atrial electrical remodeling measured by abnormal PTFV_1_ in patients with first-time AMI independent of ventricular function. Moreover, we propose PTFV_1_ as a broadly available clinical marker for increased SDB risk in cardiovascular patients.

### 4.1. Possible Mechanisms for an Abnormal PTFV_1_ in SDB

We report here a prevalence of SDB in patients with AMI of 54.5% with 25.9% central sleep apnea, which closely resembles previous data reporting an SDB prevalence ranging from 33.1% to 50% with about 20% central sleep apnea [4,20,21].

CSA in patients with heart failure is commonly explained by pulmonary congestion due to ventricular overload with consequent autonomic triggered tachypnea and subsequently reduced PaCO_2_, which results in the occurrence of an apnea episode. This leads to accumulation of PaCO_2_ and restoration of respiratory effort. However, CSA could also have pathophysiological effects on the heart that are independent of ventricular dysfunction. A small study by Lanfranchi showed that severe CSA was associated with increased arrhythmic risk without association to the severity of hemodynamic impairment due to LV dysfunction. This association may be caused by CSA-mediated nocturnal desaturations, which have been proposed as a consequence of impaired autonomic control and disturbed chemoreflex–baroreflex interactions frequently found in CSA [22].

Interestingly, for patients with AMI, a high probability of CSA-dependent nocturnal oxygen desaturations has already been shown [21]. We observe here a high ODI among patients with AMI, which strongly correlates with abnormal PTFV_1_ independent from many clinical covariates including left ventricular ejection fraction, which might provide an interesting insight into the pathogenesis of atrial remodeling and the development of atrial cardiomyopathy.

There is growing evidence that atrial structural and electrical remodeling even in the absence of atrial fibrillation can also increase the risk of clot formation and cardioembolic stroke. The latter alterations, also known as atrial cardiomyopathy, expand the traditional view of clot formation [13,23,24,25]. In fact, the ongoing ARCADIA trial is investigating the optimal anticoagulant therapy (anticoagulant therapy vs. standard ASA therapy) in patients with cryptogenic stroke and atrial cardiomyopathy and specifically uses an abnormal PTFV_1_ as an additional clinical marker for atrial cardiomyopathy [26]. We have recently shown that an abnormal PTFV_1_ is linked to increased CaMKII-dependent atrial pro-arrhythmic activity and atrial contractile dysfunction [4,19]. Atrial CaMKII is a key regulator of cardiac excitation–contraction coupling and plays an important role in triggering arrhythmias and atrial electrical remodeling [4]. Beside arrhythmias, it is tempting to speculate that CaMKII-dependent atrial contractile dysfunction may also be involved in atrial clot formation even in the absence of atrial fibrillation. Thus, CaMKII may be a promising novel treatment target for patients with atrial cardiomyopathy. In this context, the mechanisms of CaMKII activation should be elucidated in more detail. Beside the canonical Ca-dependent activation, CaMKII has been shown to be activated by increased amounts of reactive oxygen species (ROS) [27,28]. SDB-related intermittent hypoxia with consequently increased generation of ROS [29] may result in activation of atrial CaMKII and CaMKII-dependent electrical remodeling manifesting as abnormal PTFV_1,_ but this remains to be shown. Additionally, only little is known about SDB-related hypoxia and electrical atrial remodeling before atrial fibrillation emerges.

Interestingly, in patients with abnormal PTFV_1_, atrial fibrosis was less likely to be observed [19], indicating that the generation of abnormal PTFV_1_ may require functional cardiomyocytes.

Beside SDB and SDB-related hypoxia, acute myocardial infarction may also lead to acute ventricular contractile dysfunction, which could also contribute to atrial functional and/or structural alterations.

A longitudinal study recently demonstrated that increasing NT-proBNP levels were associated with LA remodeling and LA contractile dysfunction [30]. In the current study, we observed significantly higher NT-proBNP levels at discharge and lower LV EF in the group with abnormal PTFV_1_, which may contribute to impaired atrial function and abnormal PTFV_1_. In accordance, we recently demonstrated a significant negative correlation between functional LA parameters, such as LA conduit and reservoir function, as measured by feature-tracking (FT) strain analysis of cardiac magnetic resonance (CMR) images, and the extent of PTFV_1_ [19]. In contrast to atrial strain, volumetric MRI parameters for LA function such as systolic LA area or LA FAC did not show a significant association with PTFV_1_ in the present study, which agrees with previous studies [31,32].

On the other hand, multivariate linear regression analysis revealed that neither higher NT-proBNP levels nor lower LVEF were significantly associated with the magnitude of PTFV_1_ if SDB and SDB-related hypoxia were also incorporated in the multivariate model. This suggests that ventricular contractile dysfunction is unlikely to contribute decisively to the extent of PTFV_1_, at least when there is concomitant SDB.

Consistent with this, in the current study, there was also no association of PTFV_1_ with acute ischemia markers (creatine kinase, troponin I), which may correlate with infarct size and affect LV function. In addition to the possible subordinate role of LV dysfunction for PTFV_1_, an explanatory approach could also be that a proportion of patients were protected from more extensive infarct-associated ventricular myocardial injury by ischemic preconditioning due to the repetitive SDB-associated hypoxia, which has been shown previously [33]. However, the latter phenomenon should be interpreted with caution and cannot be generalized to all patients after AMI, because the healing process, as measured by myocardial salvage and reduction in infarct size, was worse in patients with SDB within three months after AMI [34]. In addition, patients with AMI and SDB showed worse hospital outcomes [21,35,36]. Regardless of a possible protective or detrimental role of SDB for ventricular injury after AMI, the role of ventricular injury for atrial remodeling and the extent of PTFV_1_ may be less important, as discussed above.

### 4.2. PTFV_1_ as a Diagnostic Marker for SDB and SDB-Related Arrhythmias

It has been found that patients with SDB especially CSA have higher severity of ACS and worse prognosis with longer hospital stay and more complications during hospitalization [21]. However, a clinical marker identifying patients at highest risk is lacking. In our cohort, oxygen desaturation index as a measure of nocturnal desaturation was significantly associated with abnormal PTFV_1_. Therefore, measurement of PTFV_1_ may be a simple and cost-effective tool for stratifying patients admitted to the hospital with a first-time AMI. Measurement of PTFV_1_ was highly reliable in different observers (Table A1). Therefore, we suggest that all patients with abnormal PTFV_1_ should receive PSG and be stratified according to their SDB risk for follow-up care.

Unfortunately, CPAP therapy may be without benefit for patients with sleep apnea [7,8,9,10], so new treatment options are urgently needed. We have recently shown that increased CaMKII activity is significantly associated with abnormal PTFV_1_ [19]. Currently, several CaMKII inhibitors are under preclinical investigation [37]. One could speculate that abnormal PTFV_1_ might help in selecting patients who could benefit from specific pharmacological treatment, such as CaMKII inhibition.

### 4.3. Limitations

This was a cross-sectional study at a single center with a relatively small sample size that was not designed to examine long-term follow-up of clinical endpoints. In addition, we do not know whether the abnormal PTFV_1_ we detected at the time of myocardial infarction is a transient phenomenon or persists over time. Larger studies are needed to validate our findings and to investigate the impact on cardiac arrhythmias and serious adverse cardiac events including heart failure exacerbations. Moreover, the definition of the negative part of the P-wave based on the isoelectric line in a slightly rising PR segment is sometimes difficult. However, the interobserver variability ICC for PTFV_1_ measurements in this study showed very good accuracy (ICC 0.888; lower CI 0.647; upper CI 0.951, Table A1).

## 5. Conclusions

This study shows that abnormal PTFV_1_ is tightly linked to SDB and especially to central instead of obstructive sleep apnea. Therefore, we hypothesize that atrial dysfunction expressed as abnormal PTFV_1_ is caused by stimulation of ROS-dependent pathways due to intermittent hypoxia represented here predominantly in CSA independent of ventricular function.

We show that the severity of SDB can be easily recognized by PTFV_1_. This ubiquitously available ECG parameter may thus be a simple and cost-effective tool to stratify patients admitted to hospital with first-time AMI for further PSG. Therefore, all patients with abnormal PTFV_1_ should obtain PSG and be stratified for follow-up care.

## Figures and Tables

**Figure 1 jcm-10-05555-f001:**
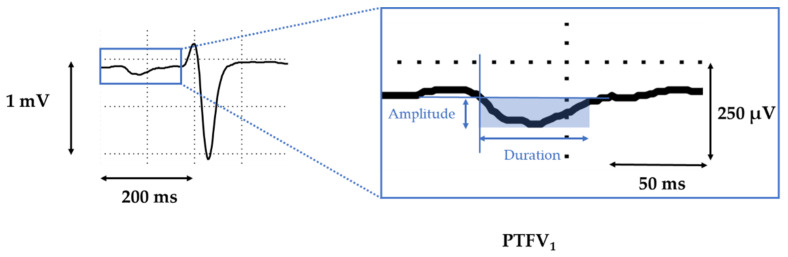
P wave terminal force in lead V_1_. Inset shows magnification.

**Figure 2 jcm-10-05555-f002:**
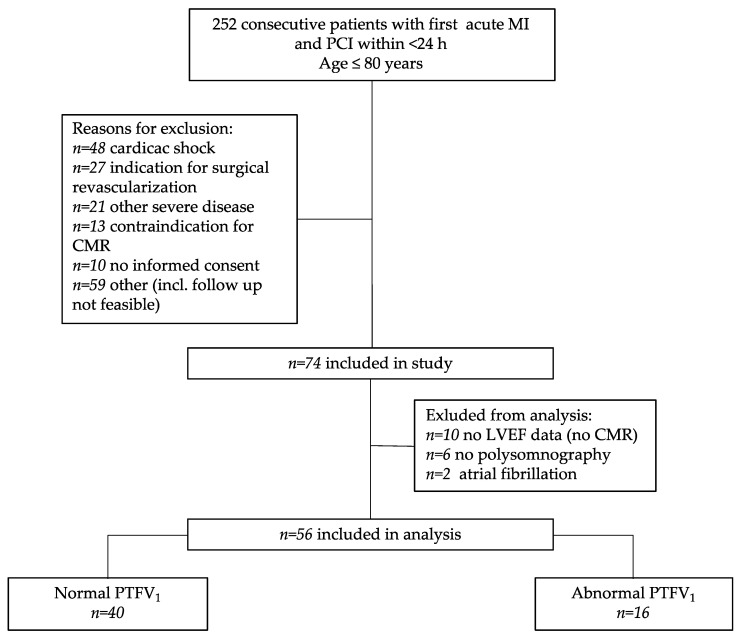
Flow diagram.

**Table 1 jcm-10-05555-t001:** Baseline characteristics: normal PTFV_1_ and abnormal PTFV_1_.

		Normal PTFV_1_ (*n* = 40)		Abnormal PTFV_1_ (*n* = 16)		
		Mean	SD	Mean	SD	*p* Value
Age	[years]	53.88	±9.87	57.88	±9.63	0.174 ^T^
BMI	[kg*m^−2^]	28.52	±3.06	28.82	±3.99	0.771 ^T^
Male	[n, %]	34 (85%)	n.a.	11 (68.8%)	n.a.	0.263 ^Chi^
Arterial hypertension	[n, %]	19 (47.5%)	n.a.	9 (60%)	n.a.	0.409 ^F^
Diabetes mellitus	[n, %]	6 (15%)	n.a.	3 (20%)	n.a.	0.692 ^F^
Hypercholesterolemia	[n, %]	12 (30%)	n.a.	5 (33.3%)	n.a.	1.000 ^F^
LDL-cholesterol	[mg*dL^−1^]	136.53	±35.33	111.57	±23.4	0.018 ^T^
Smoking	[n, %]	30 (75%)	n.a.	11 (73.3%)	n.a.	1.000 ^F^
SDB	[n, %]	17 (42.5%)	n.a.	13 (86.7%)	n.a.	**0.003** ^ **Chi** ^
STEMI	[n, %]	37 (92.5%)	n.a.	11 (68.8%)	n.a.	**0.035** ^ **F** ^
CK max	[U*L^−1^]	1993.49	±1393.21	2232.07	±1588.63	0.590 ^T^
Troponin I max	[ng*mL^−1^]	29.11	±66.63	40.26	±90.6	0.638 ^T^
NT-proBNP at discharge	[pg*mL^−1^]	774.47	±835.61	2201.19	±1390.37	**0.002** ^ **W** ^
eGFR	[mL*min^−1^*1, 73 m^−2^]	95.16	±16.53	83.63	±28.03	0.152 ^W^
Resting heart rate	[min^−1^]	75.46	±12.13	75.33	±22.14	0.983 ^W^
Systolic blood pressure	[mmHg]	127.43	±22.79	127.67	±17.65	0.971 ^T^
Diastolic blood pressure	[mmHg]	78.43	±12.95	75.8	±11.38	0.493 ^T^
LV EF	[%]	48.93	±7.45	43.15	±11.51	**0.035** ^ **T** ^
RV EF	[%]	58.25	±8.98	59	±11.24	0.808 ^T^
TAPSE	[mm]	20.12	±6.01	19.99	±4.33	0.943 ^T^
Systolic LA area	[cm^2^]	25.9	±4.19	24.67	±3.53	0.369 ^T^
Diastolic LA area	[cm^2^]	18.11	±3.03	18.44	±3.82	0.764 ^T^
LA FAC	[%]	32.56	±8.41	30.75	±11.45	0.574 ^T^
ACEi/ARB at discharge	[n, %]	38 (97.4%)	n.a.	15 (100%)	n.a.	1.000 ^F^
ACEi/ARB at admission	[n, %]	4 (10%)	n.a.	1 (6.7%)	n.a.	1.000 ^F^
β-Blocker at discharge	[n, %]	37 (97.4%)	n.a.	14 (93.3%)	n.a.	0.490 ^F^
β-Blocker at admission	[n, %]	1 (2.5%)	n.a.	1 (6.7%)	n.a.	0.475 ^F^
Loop diuretics at discharge	[n, %]	14 (36.8%)	n.a.	8 (53.3%)	n.a.	0.272 ^Chi^
Loop diuretics at admission	[n, %]	0	n.a.	0	n.a.	n.a.
MRA at discharge	[n, %]	16 (42.1%)	n.a.	10 (66.7%)	n.a.	0.107 ^Chi^
MRA at admission	[n, %]	0	n.a.	0	n.a.	n.a.

ACEi: ACE-inhibitor; ARB: angiotensin receptor blocker; AHI: apnea-hypopnea-index; BMI: body mass index; CK: creatine kinase; EF: ejection fraction; eGFR: estimated glomerular filtration rate; FAC: fractional area change; LA: left atrium; LV: left ventricle; NT-proBNP: N-terminal pro-B-type natriuretic peptide; MRA: Mineralocorticoid receptor antagonist; PTFV_1_: P wave terminal force in lead (abnormal ≥4000 µV*ms); RV: right ventricle; SD: standard deviation; SDB: sleep-disordered breathing; STEMI: ST-elevation myocardial infarction; TAPSE: tricuspid annular plane systolic excursion. Bold values mean statistical significance calculated by the two-sided Student‘s *t*-test(^T^), Welch’s *t*-test(^W^), chi-square test(^Chi^) or Fischer´s exact test(^F^).

**Table 2 jcm-10-05555-t002:** Respiratory and sleep characteristics.

		Normal PTFV_1_ (*n* = 40)		Abnormal PTFV_1_ (*n* = 16)		
		Mean	SD	Mean	SD	*p* Value
SDB	[n, %]	17 (42.5%)	n.a.	13 (86.7%)	n.a.	**0.003** ^ **Chi** ^
-OSA	[n, %]	10 (25.6%)	n.a.	6 (40%)	n.a.	0.333 ^F^
-CSA	[n, %]	7 (17.9%)	n.a.	7 (46.7%)	n.a.	**0.043** ^ **F** ^
AHI	[h-1]	14.64	±13.91	36.14	±24.87	**<0.001** ^ **T** ^
oAHI	[h-1]	8.10	±8.16	12.82	±10.43	0.084 ^T^
cAHI	[h-1]	6.75	±9.55	23.32	±27.03	**0.034** ^ **W** ^
ODI	[h-1]	11.39	±9.88	28.77	±23.69	**0.018** ^ **W** ^
SaO_2_ mean	%	93.18	±2.26	93.00	±1.73	0.783 ^T^
SaO_2_ min	%	85.74	±5.87	82.20	±6.09	0.055 ^T^
Sleep efficiency	%	72.15	±16.25	69.95	±12.77	0.653 ^T^
REM	%	16.07	±6.17	14.13	±7.23	0.327 ^T^
ESS		7.32	±4.57	5.75	±2.60	0.147 ^W^

AHI: apnea-hypopnea-index; CSA: central sleep apnea; ESS: Epworth Sleepiness Scale score; ODI: oxygen desaturation index; OSA: obstructive sleep apnea; PTFV_1_: P wave terminal force in lead (abnormal ≥4000 µV*ms); REM: % of total sleep time spent in rapid eye movement sleep stage; SD: standard deviation; SaO_2_: arterial oxygen saturation; SDB: sleep-disordered breathing; Bold values mean statistical significance calculated by the two-sided Student‘s *t*-test(^T^), Welch´s *t*-test(^W^), chi-square test(^Chi^) or Fischer´s exact test(^F^).

**Table 3 jcm-10-05555-t003:** Univariate linear regression of PTFV_1_.

	Univariate Linear Regression Analysis with PTFV_1_			
PTFV_1_ [µV*ms]	B	95% CI	R^2^ (adj.)	*p* Value
ODI [h-1]	68.116	35.992 to 100.240	0.268	**<0.001**
AHI [h-1]	48.845	22.644 to 75.045	0.197	**<0.001**
cAHI [h-1]	46.810	15.375 to 78.246	0.128	**0.004**
oAHI [h-1]	56.127	−7.940 to 120.194	0.039	0.085
NT-proBNP at discharge [pg/mL]	0.628	0.109 to 1.148	0.097	**0.019**
LV EF [%]	−60.863	−132.377 to 10.651	0.036	0.094
Age [y]	47.404	−9.450 to 104.258	0.032	0.100
eGFR [mL*min^−1^* 1,73 m^−2^]	−23.099	−51.033 to 4.845	0.032	0.103
RR sys [mmHg]	19.564	−8.311 to 47.438	0.018	0.165
BMI [kg/m^2^]	87.580	−88.285 to 263.446	< 0.001	0.322
Trop I max [ng/mL]	3.950	−4.516 to 12.416	−0.002	0.353
Systolic LA area	−71.394	−239.042 to 96.255	−0.006	0.395
CK max [U/l]	0.132	−0.276 to 0.540	−0.011	0.518
Smoking	402.290	−943.843 to 1748.422	−0.012	0.551
Male sex	−303.380	−1741.292 to 1134.532	−0.015	0.674
Diabetes mellitus	251.791	−1296.227 to 1799.809	−0.017	0.745
LA FAC [%]	−10.620	−85.278 to 64.038	−0.022	0.775

AHI: apnea-hypopnea-index; BMI: body mass index; BNP: brain natriuretic peptide; CI: confidence interval; CK: creatine kinase; EF: ejection fraction; eGFR: estimated glomerular filtration rate; FAC: fractional area change; LA: left atrium; LV: left ventricle; ODI: oxygen desaturation index; PTFV_1_: P wave terminal force in lead V_1_; RA: right atrium; RRsys: systolic blood pressure; Trop: Troponin I; Bold values mean statistical significance.

**Table 4 jcm-10-05555-t004:** Multivariate linear regression of PTFV_1_.

	Model 1 (with ODI) Multiple Linear Regression Analysis R^2^ = 0.408 (adj. R^2^ = 0.317); *p* = 0.002		Model 2 (with AHI) Multiple Linear Regression Analysis R^2^ = 0.330 (adj. R^2^ = 0.227); *p* = 0.012		Model 3 (with cAHI) Multiple Linear Regression Analysis R^2^ = 0.256 (adj. R^2^ = 0.186); *p* = 0.014	
PTFV_1_ [µV*ms]	B ^*^ [95% CI]	P ^#^	B ^*^ [95% CI]	P ^#^	B ^*^ [95% CI]	P ^#^
ODI [h-1]	65.619 [29.717 to 101.522]	0.001				
AHI [h-1]			45.170 [11.903 to 78.437]	**0.009**		
cAHI [h-1]					45.172 [11.905 to 78.440]	**0.009**
oAHI [h-1]	−20.049 [−87.368 to 47.269]	0.550	−7.992 [−79.286 to 63.303]	0.822	37.178 [−27.626 to 101.983]	0.253
NT-proBNP at discharge [pg/mL]	0.375 [−0.139 to 0.888]	0.148	0.402 [−0.146 to 0.950]	0.146	0.402 [−0.146 to 0.950]	0.146
LV EF [%]	−60.432 [−128.593 to 7.729]	0.081	−50.472 [−122.825 to 21.882]	0.166	−50.472 [−122.825 to 21.881]	0.166
Age [y]	−24.189 [−101.905 to 53.527]	0.533	−40.920 [−123.645 to 41.806]	0.323	−40.917 [−123.642 to 41.808]	0.323
eGFR [mL*min^−1^* 1,73 m^−2^]	−24.263 [−58.700 to 10.175]	0.162	−30.712 [−67.833 to 6.409]	0.102	−30.710 [−67.831 to 6.411]	0.102

AHI: apnea-hypopnea-index; CI: confidence interval; EF: ejection fraction; eGFR: estimated glomerular filtration rate; LV: left ventricle; NT-proBNP: N-terminal pro-B-type natriuretic peptide; ODI: oxygen desaturation index; PTFV_1_: P wave terminal force in lead V_1_; Bold values mean statistical significance, * beta coefficient, # *p* value.

**Table 5 jcm-10-05555-t005:** Univariate and multivariate linear regression of AHI.

	Univariate Linear Regression Analysis with AHI			Model 1 Multiple Linear Regression Analysis R^2^ = 0.326 (adj. R^2^ = 0.213); *p* = 0.021		Model 2 Multiple Linear Regression Analysis R^2^ = 0.351 (adj. R^2^ = 0.245); *p* = 0.010	
AHI [h-1]	B [95% CI]	*p* Value	R^2^ (adj.)	B [95% CI]	*p* Value	B [95% CI]	*p* Value
PTFV_1_ [µV*ms]	0.004 [0.002 to 0.07]	**<0.001**	0.197	0.004 [0.001 to 0.07]	**0.024**		
Abnormal PTFV_1_	21.495 [10.872 to 32.118]	**<0.001**	0.223			21.209 [4.452 to 37.966]	**0.015**
BMI [kg/m^2^]	1.807 [0.231 to 3.382]	**0.025**	0.074	1.737 [−0.156 to 3.630]	0.071	1.500 [−0.296 to 3.295]	0.099
NT-proBNP at discharge [pg/mL]	0.005 [<0.001 to 0.009]	0.062	0.053	0.002 [−0.004 to 0.007]	0.487	0.001 [−0.007 to 0.006]	0.823
Systolic LA area	0.876 [−0.187 to 1.940]	0.104	0.038	−0.048 [−1.187 to 1.091]	0.932	0.255 [−0.815 to 1.325]	0.632
Smoking	−8.729 [−20.929 to 3.471]	0.157	0.019	−7.268 [−21.764 to7.227]	0.316	−9.148 [−23.060 to 4.765]	0.191
LV EF [%]	−0.443 [−1.075 to 0.188]	0.164	0.020	−0.125 [−0.832 to 0.582]	0.723	−0.148 [−0.798 to 0.501]	0.646
Male sex	7.269 [−6.631 to 21.169]	0.299	0.002				
LA FAC [%]	−0.299 [−0.985 to 0.386]	0.384	−0.005				
Age [y]	0.218 [−0.342 to 0.778]	0.438	−0.007				
RR sys [mmHg]	0.073 [−0.182 to 0.328]	0.570	−0.013				
Trop I max [ng/mL]	0.012 [−0.068 to 0.092]	0.768	−0.019				
eGFR [mL*min^−1^* 1,73 m^−2^]	−0.030 [−0.294 to 0.234]	0.822	−0.018				
CK max [U/l]	<0.001 [−0.005 to 0.005]	0.917	−0.021				
Diabetes mellitus	0.416 [−14.225 to 15.057]	0.955	−0.019				

AHI: apnea-hypopnea-index; BMI: body mass index; CI: confidence interval; CK: creatine kinase; EF: ejection fraction; FAC: fractional area change; LA: left atrium; LV: left ventricle; NT-proBNP: N-terminal pro-B-type natriuretic peptide; ODI: oxygen desaturation index; P wave terminal force in lead V_1_ (PTFV_1_) (abnormal ≥4000 µV*ms); RA: right atrium; RRsys: systolic blood pressure; Trop: Troponin I; Bold values mean statistical significance.

## Data Availability

The data presented in this study will be shared on reasonable request to the corresponding author. The data are not publicly available due to privacy restrictions.

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
