# Peer review of "Central Sleep Apnea Is Associated with an Abnormal P-Wave Terminal Force in Lead V1 in Patients with Acute Myocardial Infarction Independent from Ventricular Function"

_jcm, 2021, doi:10.3390/jcm10235555_

Round 1

Reviewer 1 Report

This is a very good writing format style and scientific methods article. Here are comments that further need to be clarified. 

1 . Atrial proarrhythmic activity is related to PTFV1 as authors mentioned. Is this EKG parameter also associated with atrial cardiomyopathy? Could you please further discuss about the relationship/association between atrial proarrhythmic activity and ACM in the light of rising popularity in ACM concept since 2016?

2. Absolutely, PTFV1 is related to atrial pathology, do you also analyze any other p wave parameters that involves the atrial health status such as interatrial block? As authors provided only less than a hundred case, further explore these findings.

In conclusion, we praise the authors for conducting this clinical meaningful article involving from basic electrophysiology to clinical use.  

Reviewer 2 Report

Reviewer comment:

In this study, Jan Pec et al sought to this study investigates the relationship between Sleep-disordered breathing (SDB) and SDB-related hypoxia with the P wave terminal force in lead V1 (PTFV1) in patients presenting with acute myocardial infarction. The authors found a significant association between SDB-related hypoxia and the magnitude of PTFV1 independent from other relevant clinical co-factors in a multivariable regression analysis, and concluded that the PTFV1 may be used as a clinical marker for increased SDB risk in cardiovascular patients. Although this review provided important clinical perspectives on the important issue, there exists several limitations.

 Major comments:

#1.         If abnormal PTFV1 is associated with atrial proarrhythmic activity due to the stimulation of reactive oxygen-species-dependent pathways, as described in the abstract, patients with abnormal PTFV1 may have had higher incidence of new-onset atrial tachyarrhythmias and/or major adverse cardiac events including heart failure exacerbation than those with normal PTFV1 during long-term follow-up, which should be further investigated.

#2.         Whether abnormal PTFV1 persists or not during a course of acute myocardial infarction remains unknown. The serial changes in PTFV1 may distinguish transient abnormal PTFV1 from persistent abnormal PTFV1, possibly associated with the long-term impact on the incidence of new-onset atrial tachyarrhythmias.

#3.        The authors stated that treatment of sleep apnea using continuous positive airway pressure (CPAP), or adaptive servo-ventilation failed to show significant benefit regarding cardiovascular outcomes. Currently, it remains uncertain whether early detection of SDB improve clinical outcomes in patients with acute myocardial infarction. Therefore, the authors should clarify the clinical implications of this study.

#4.         Manual measurement of PTFV1 highly depends on the observer’s skill or experience, though intra class correlation showed acceptable inter-observer reproducibility in this study. It is helpful for readers to demonstrate the actual measurement of the PTFV1 in Figure 1.

Minor comments:

#1.         The first sentence of 3.3, ‘A readily available marker such as PTFV1 to identify patients with an increased prevalence of SDB may be hugely helpful in clinical routine to identify patients for PSG. We therefore tested if the association of PTFV1 with SDB may be used as a screening method in patients presenting with acute myocardial infarction.’ should be described in the methods section.

#2.         Figure 3 needs revision, as the findings provided by this figure is unclear.

Round 2

Reviewer 2 Report

The authors provided sincere response and revision according to the previous comments. I have no further comments.